# Evaluation of morphological variations of mandibular bone in adult bruxers using CBCT: A cross-sectional study

Estelle Casazza[1]*, Benoit Ballester[2], Clémence Vernet[3], Camille Philip-Alliez[4], Anne Raskin[1]

**1** Aix Marseille University, APHM, CNRS, EFS, ADES, Hôpital de la Timone, Pôle PROMOD ODONTO, Service d'Odontologie Hospitalière et Chirurgie orale, Marseille, France, **2** Aix Marseille University, APHM, INSERM, IRD, SESSTIM, Sciences Economiques & Sociales de la Santé & Traitement de l'Information Médicale, ISSPAM, Hôpital de la Timone, Pôle PROMOD ODONTO, Service de Réhabilitations Orales, Marseille, France, **3** Dental Private Practice, Marseille, France, **4** Aix Marseille University, APHM, Gustave Eiffel University, LBA, Hôpital de la Timone, Pôle PROMOD ODONTO, Service d'Orthopédie Dento-Faciale et d'Odontologie Pédiatrique Générale, Marseille, France

* estelle.casazza@ap-hm.fr

## Abstract

This cross-sectional study aimed to establish whether a difference exists between the mandibular bone density and the mandibular angle values of adult bruxer and non-bruxer patients, based on a CBCT analysis. CBCT scans of bruxer and non-bruxer patients were analysed with two software packages, 3D Slicer® and Romexis®. Bone density in the alveolar bone below and immediately adjacent to the apices of teeth 31, 33, 34, 41, 43, 44, and the mesial apices of teeth 36 and 46, was recorded in Hounsfield units with 3D Slicer®. The mandibular angle between the corpus and ramus tangent lines was measured using Romexis® software. 113 CBCT scans were included in the study of alveolar bone density, of which 78 were used to study mandibular angle values. A statistically significant difference ($p < 0.05$) was noted between the density values of the two groups, with higher values in bruxers than in non-bruxers. Mandibular angle values were significantly lower in bruxers ($p < 0.05$). This cross-sectional study based on CBCT imaging showed certain morphological differences between the mandibles of bruxers and those of non-bruxers. Further studies are needed to supplement this preliminary research, in particular prospective studies.

## Introduction

The mandible, the only movable bone in the human skull, has a horseshoe-shaped body (corpus) that connects to two ascending rami at the mandibular angles. Like all skeletal bones, the mandible undergoes constant remodelling under the influence of various factors. These may be systemic, for example hormones or growth factors, or local, for example the mechanical stresses generated by the action of the muscles

**Data availability statement:** All relevant data are within the manuscript and its Supporting information files.

**Funding:** The author(s) received no specific funding for this work.

**Competing interests:** The authors have declared that no competing interests exist.

of the masticatory apparatus [1–3]. Several muscles that attach to the mandible serve essential functions such as mastication, phonation, swallowing, and emotional management. The most powerful muscles are the mandibular elevator muscles (masseter, temporalis, medial pterygoid) and the muscle responsible for protrusion of the mandible (lateral pterygoid) [4]. These muscles attach to the mandible at various sites, so connecting it to the craniofacial skeleton. The interaction between bone tissue and the muscular system, which applies mechanical stresses to the bone, is crucial in maintaining the balance of bone remodelling [5]. This bone remodelling can manifest as variations in various bone tissue characteristics, such as volume or density [6]. Moreover, bone variability is dependent on two types of characteristic: innate, transmitted via genetic inheritance, and acquired, developed through behaviour. Bruxism belongs to the latter category.

In 2024, unspecified bruxism was defined as "a repetitive jaw-muscle activity characterised by clenching or grinding of the teeth and/or by bracing or thrusting of the mandible" [7]. When excessive, this mandibular behaviour, thought to be exhibited by nearly 22% of the adult population globally [8], can weaken the teeth and the temporomandibular joints, and generate mastication myalgia [9]. Bruxism occurs in affected subjects while awake (with repetitive or sustained tooth contact and/or specific mandibular movements) or asleep (with rhythmic or non-rhythmic masticatory muscle activity), and can generate interdental contact through clenching or grinding [7]. The force developed by bruxer patients can be up to three times greater than that exerted during the functional activity of the masticatory apparatus [10]. These forces, which are more intense and occur more frequently than those observed in ordinary mastication, are transmitted not only to the teeth but also to the supporting dental tissues, notably alveolar bone [11], with different types of impact on these structures [12]. These mechanical loads can lead to architectural changes in mandibular bone tissue, as Julius Wolff pointed out as early as 1892, when he asserted that "bone in a healthy person or animal adapts to the forces to which it is subjected". Thus bone can modify its external cortical and trabecular structure, according to the nature of the forces exerted [13]. When an adequate mechanical stress is applied to bone, bone mass can increase. However, excessive stress can lead to increased bone resorption, with the risk of fatigue fracture due to the formation of incompetent bone. Conversely, bone that is unstressed, as is the case with edentulous patients exhibits a high level of resorption associated with reduced bone formation.

The study of mandibular morphology in living subjects requires the use of medical imaging. Several types of radiological examination can be performed as part of a routine dental check-up: panoramic radiography, a two-dimensional imaging examination, and CBCT (Cone Beam Computed Tomography), a three-dimensional examination. CBCT is a medical imaging technique that offers many advantages: three-dimensional assessment of mandibular hard tissue, with no projection or superimposition of anatomical structures, a high level of accuracy and reliability, three-dimensional reconstruction, combined with a radiation dose that is often lower than that of other three-dimensional imaging techniques such as CT scans [14]. Panoramic radiography has been used in several studies to evaluate various parameters in different mandibular

areas in bruxer patients [15–20], but the radiological indices used and the areas of interest varied greatly from one study to another, making comparisons of results difficult. Furthermore, the data obtained were relative, as they didn't allow the expression of volume or density. This is not the case for CBCT, which to our knowledge has only been used in one study [21]. The aim of this cross-sectional study was to evaluate the presence of morphological variations of mandibular bone in adult bruxers and non-bruxers using CBCT, using two markers: alveolar bone density and mandibular angle value.

## Materials and methods

This study followed the recommendations of the STROBE Statement checklist (see S1 Appendix).

This retrospective cross-sectional clinical study was based on usable mandibular CBCTs performed in adult patients seen in consultation in the Odontology and Oral Surgery Department of the PROMOD-ODONTO Center of Timone Hospital, AP-HM (Assistance Publique – Hôpitaux de Marseille), France, diagnosed as bruxers or non-bruxers with a self-administered clinical examination over a three-year period (January 2021-January 2024). Information concerning date of birth, gender, and bruxer/non-bruxer status was collected on 4 April 2024 for each patient from their computerised medical records.

Authorisation was obtained from the AP-HM Data Protection Officer (AP-HM Health Research Division) for collection and processing of the data, which were anonymised (recorded in the RGPD-APHM register under number 2022−21). In line with current regulations, patients were informed of the study, and consent was not required.

### Participants

**Subject inclusion criteria.**

- adult patients, of both genders;

- dentate patients, with no more than two missing teeth on the dental arch, or in whom missing teeth had been replaced by an implant (excluding the third molars);

- patients diagnosed as bruxers or non-bruxers, based on clinical signs mentioned in the computerised medical records, such as tooth wear, masseter hypertrophy, tongue scalloping, and alveolar bone exostosis;

- patients who had undergone a wide-field CBCT imaging examination of the whole mandible, performed solely within the PROMOD-ODONTO Centre by the same medical radiology technician.

**Criteria for non-inclusion of subjects.**

- patients with partial or total mandibular edentulism;

- patients with partial or total maxillary edentulism, not restored by a fixed, tooth-borne, or implant-borne prosthesis;

- patients who had undergone a small field-of view CBCT, or a CBCT centred on the maxilla, or performed in a facility other than the Dentistry and Oral Surgery Department (PROMOD-ODONTO Center) of Timone Hospital;

- patients with localised pathologies of the mandible, such as bone metastases, osteoporosis, or a hormone deficiency (parathyroid hormone or calcitonin), patients with a history of mandibular fracture, or who had undergone radiotherapy in the oral region, or with mandibular asymmetry;

- patients with periapical inflammation.

### Sample size calculation

The sample size required for this study was calculated with a clinically relevant minimum difference between groups of 200 Hounsfield units (HU), an estimated standard deviation of 250 HU [22], a significance level of $\alpha = 0.05$ and statistical

power of 90%.The sample size required for this study was 66 CBCTs. 113 CBCTs were used to compensate for any missing data (missing tooth, missing mandibular angle for instance).

## CBCT

All patient CBCTs were performed on the same X-ray machine: Planmeca ProMax®3D Mid (Planmeca OY, Helsinki, Finland), with known acquisition parameters (90 kV, 13mA, and 15s acquisition time, 0.45 mm slices, 150μm resolution, 100 mm diameter, maximum contrast) by a single operator. All CBCT images were exported in DICOM format.

**CBCT analysis: Study of bone density values.** All CBCT scans were analysed with 3D Slicer®, version 5.6.1, an open-access software package for processing data as DICOM data [23]. Imaging examinations were evaluated by two blinded operators, EC and BB. Values were recorded in Hounsfield units (HU) using a 1.5 mm diameter spherical selection tool in the alveolar bone below and immediately adjacent to the apices of teeth 31, 33, 34, 41, 43, 44, and the mesial apices of teeth 36 and 46. If a tooth was missing, no value was recorded. The correct location of the volume analysed was checked using the three planes available simultaneously (sagittal, frontal, horizontal). This volume had to contain alveolar bone only, with no dental tissue, periodontal ligament, cortical bone, or anatomical elements such as the inferior alveolar nerve. An example of zone selection for each dental class (Fig 1).

**CBCT analysis: Study of mandibular angle values.** CBCTs with an intact mandibular angle were analysed using Romexis® software, version 6.0 (Planmeca OY, Helsinki, Finland). Mandibular angle value was determined using the "Angle measurement" tool available in the "Annotation" tab, based on the sagittal view of the right side of each mandible.

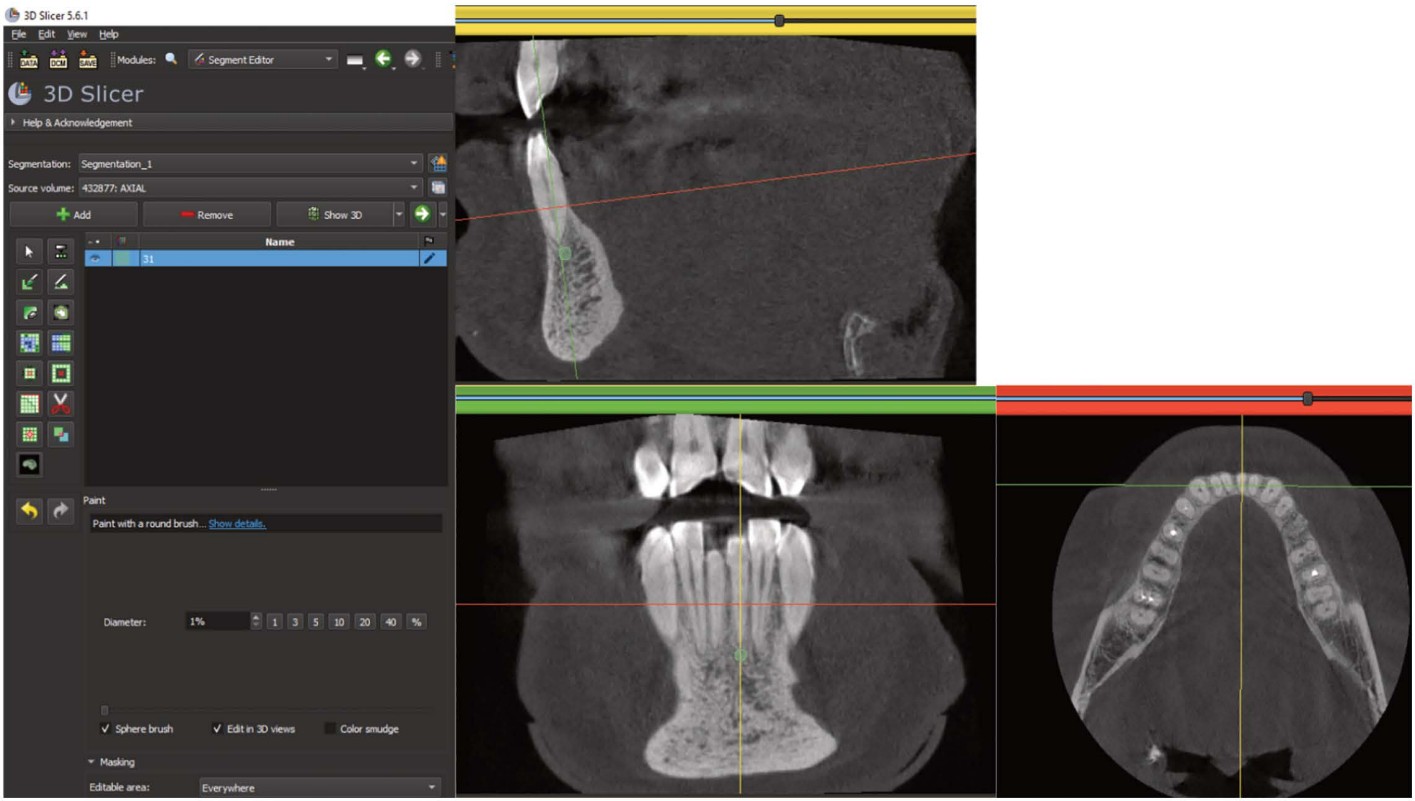

**Fig 1. Selection of the density zone (green sphere) below the apex of tooth 31 in sagittal section (yellow band), with verification in coronal view (green band).** Tooth position was checked in horizontal section (red band).

The mandibular angle corresponded to the angle (expressed in degrees) formed by the corpus tangent line and the ramus tangent line, as described by Schudy's analysis [24]. This angle was displayed on the three-dimensional reconstruction of the mandible under investigation, enabling verification of the correct positioning of the tangents in space and validation of the mandibular angle measurement indicated by the software.

### Statistical analysis

Analyses were performed using R software (version 4.0.3).

**Intra-operator variability and inter-operator variability for bone density values.** The CBCTs were evaluated by two blinded operators, EC and BB. Intra-operator repeatability and inter-operator reproducibility were assessed for the 8 quantitative measurements with Bland-Altman agreement tests, using 10 CBCTs selected randomly from the sample (p < 0.05).

**Intra-operator variability and inter-operator variability for mandibular angle values.** CBCT images showing intact mandibular angles were evaluated by two blinded operators, EC and CV. Intra-operator repeatability and inter-operator reproducibility were assessed for mandibular angle measurements with Bland-Altman agreement tests, using 10 CBCTs selected randomly from the sample (p < 0.05).

**Sample analysis.** The F-test for the equality of two variances was used to verify normal distribution and equality of variances between the two groups. An analysis of the whole sample was carried out with Welch's t-test (p < 0.05) to study:

- the difference in density values (in Hounsfield units);

- the difference in mandibular angle values (in degrees);

between the bruxer and non-bruxer groups.

## Results

The statistical dataset is available as supplementary material (S1–S5 Tables for bone density, S6 and S7 Tables for mandibular angle).

### Alveolar bone density

**Sample composition.** One hundred and thirteen CBCTs, representing the 113 patients who met the inclusion criteria, were thus evaluated for this study (45 bruxers and 68 non-bruxers), as shown in Table 1.

The mean age of bruxers was 31 years, with a standard deviation of 14.9 years (range: [20–66] years).

The mean age of non-bruxers was 33 years, with a standard deviation of 14.6 years (range: [19–72] years).

The cut-off point of 45 years used in Table 1 was chosen to reflect the median age of patients included in the study.

**Operator calibration.** There was a good agreement between all the quantitative measurements made by the two operators (Bland-Altman tests), both in terms of intra-operator repeatability and inter-operator reproducibility (Fig 2).

**Table 1. Distribution of patients by status (bruxer or non-bruxer), gender, and age group.**

|  |  | Status | | Total |
|---|---|---|---|---|
|  |  | Bruxer | Non-bruxer |  |
| Male | < 45 years old | 23 | 26 | 63 |
|  | > 45 years old | 4 | 10 |  |
| Female | < 45 years old | 16 | 28 | 50 |
|  | > 45 years old | 2 | 4 |  |
| Total |  | 45 | 68 | 113 |

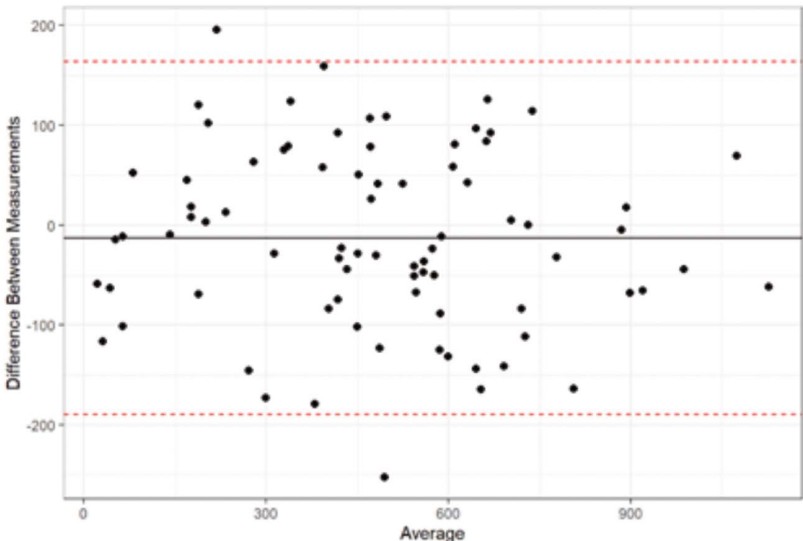

**Fig 2. Good inter-operator agreement of 8 measurements performed on 10 CBCTs selected randomly (Bland-Altman test).**

**Sample analysis.** Eight density measurements were taken for each mandible from the alveolar bone adjacent to the apical zones of teeth 31, 33, 34, 41, 43, 44, and from the mesial apices of teeth 36 and 46. The characteristics of the 8 sites recorded for the entire sample (113 CBCTs) are presented in Table 2.

It thus appears that the mandibular first molars are the teeth least represented in this research, irrespective of the side studied.

The conditions for the application of Welch's t-test were checked for bruxer and non-bruxer groups using an F-test for equality of variance between the two groups, and the normality of the distribution was verified using the Quantile-Quantile graphical method ("Q-Q plot").

The statistical analysis of the overall sample with Welch's t-test, to investigate correlations between the density values measured at 8 sites of interest and the patient's bruxer or non-bruxer status, is presented in Table 3.

The results show a statistically significant difference in density values expressed in Hounsfield units between the bruxer and non-bruxer groups (Welch's t-test, p = 0.02492).

**Table 2. Descriptive statistics for the overall sample.**

| ROI | Minimum (HU) | Q1 (HU) | Median (HU) | Mean (HU) | Standard deviation (HU) | Q3 (HU) | Maximum (HU) | Missing data (percentage and number) |
|---|---|---|---|---|---|---|---|---|
| 31 | 110.7 | 510.9 | 638.1 | 651.5 | 206.09 | 778.2 | 1109.9 | 2 (1.7%) |
| 33 | 37.09 | 377.15 | 528.34 | 540.32 | 241.86 | 689.88 | 1328.41 | 0 |
| 34 | 58.61 | 305.65 | 423.09 | 445.04 | 200.32 | 572.62 | 1000.26 | 2 (1.7%) |
| 36 | −91.95 | 108.78 | 197.97 | 276.21 | 243.83 | 433.55 | 1039.16 | 6 (5.3%) |
| 41 | 182.6 | 495.4 | 659.8 | 652.1 | 213.23 | 781.3 | 1292 | 2 (1.7%) |
| 43 | 25.49 | 424.04 | 547.55 | 567.29 | 222.08 | 666.61 | 1281.52 | 1 (0.8%) |
| 44 | −8.181 | 312.638 | 437.639 | 456.329 | 205.2 | 538.616 | 1190 | 1 (0.8%) |
| 46 | 10.42 | 140.62 | 211.46 | 277.77 | 221.45 | 355.35 | 1322.1 | 4 (3.5%) |

ROI: Region Of Interest. Q1: first quartile. Q3: third quartile. HU: Hounsfield units.

**Table 3. Statistical analysis of mandibular density measurements taken at eight sites across the entire sample as a function of patients' bruxer or non-bruxer status.**

| ROI | Mean NB Group ± SD (HU) | Mean B Group ± SD (HU) | Welch's t-test (*p-value*) | 95% CI |
|---|---|---|---|---|
| 31 | 618.3±223.85 | 703.9±163.41 | 0.02* | −158.65348 −12.54617 |
| 33 | 512.3±225.37 | 582.73±261.74 | 0.146 | −165.2038 24.2480 |
| 34 | 411.59±196.42 | 494.1±197.94 | 0.03* | −158.251309 −6.766356 |
| 36 | 261.76±248.14 | 296.91±238.82 | 0.46 | −129.81857 59.50806 |
| 41 | 622.5±199.40 | 697.2±227.70 | 0.07 | −158.368906 9.060107 |
| 43 | 530.76±220.52 | 621.68±215.47 | 0.032* | −174.13840 −7.70549 |
| 44 | 423.37±190.72 | 507.26±218.29 | 0.04* | −163.890564 −3.886432 |
| 46 | 281.55±235.26 | 272.18±201.84 | 0.82 | −74.25902 93.00538 |

ROI: Region Of Interest. NB: Non-bruxer. B: Bruxer. SD: Standard Deviation. HU: Hounsfield Units. 95% CI: 95% confidence interval. p: significance level; *p<0.05; ** p<0.001.

The mean density values expressed in Hounsfield units were higher in the bruxer group than in the non-bruxer group for all 8 sites studied, with statistically significant differences found for half of the sites.

## Mandibular angles

**Sample composition.** Of the 113 CBCTs initially included, only 78 were retained for this analysis. 35 CBCTs were excluded because they did not show the mandibular angle.

The remaining CBCTs corresponded to 38 bruxers and 40 non-bruxers. Patient distribution by gender, age and status (bruxer or non-bruxer) is shown in Table 4.

The mean age of non-bruxer patients was 26 years, with a standard deviation of 7.43 years (range: [19–54] years).

The mean age of bruxer patients was 31 years, with a standard deviation of 10.17 years (range: [21–60] years).

**Operator calibration.** There was good agreement between all the quantitative measurements made by the two operators (Bland-Altman tests), both in terms of intra-operator repeatability and inter-operator reproducibility (Fig 3).

**Sample analysis.** The descriptive parameters of the two groups of bruxer and non-bruxer patients are detailed in Table 5.

The conditions for the application of Welch's t-test were checked for each group using an F-test for equality of variance between the two groups, and the normality of the distribution was verified using the Quantile-Quantile graphical method ("Q-Q plot").

The results showed a statistically significant difference in mandibular angle values between the bruxer and non-bruxer groups (Welch's t-test, p=0.01157).

**Table 4. Patient distribution according to status (bruxer or non-bruxer), gender, and age group.**

| | Status | | |
|---|---|---|---|
| | **Bruxer** | **Non-bruxer** | **Total** |
| Male > 45 years old | 3 | 0 | 3 |
| Male < 45 years old | 24 | 13 | 37 |
| Female > 45 years old | 1 | 2 | 3 |
| Female < 45 years old | 10 | 25 | 35 |
| Total | 38 | 40 | 78 |

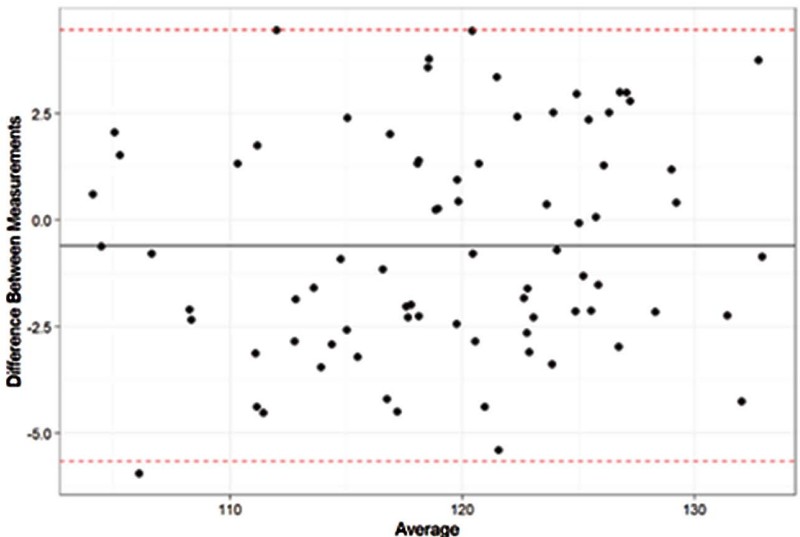

**Fig 3. Good inter-operator agreement of mandibular angle measurements for the 78 CBCTs (Bland-Altman test).** Differences between the two measurements are expressed in degrees on the y-axis.

**Table 5. Descriptive statistics for each patient group (78 mandibles).**

| Patient status | Minimum (°) | Q1 (°) | Median (°) | Mean (°) | Standard deviation (°) | Q3 (°) | Maximum (°) |
|---|---|---|---|---|---|---|---|
| Non bruxers | 108.3 | 117.6 | 122.5 | 121.3 | 5.73 | 125.5 | 132.9 |
| Bruxers | 104.1 | 111.2 | 117.9 | 117.3 | 7.78 | 122.5 | 132.7 |

Q1: first quartile. Q3: third quartile.

Mandibular angle values were lower in the bruxer group (mean 117.27°) than in the non-bruxer group (121.30°), with a 95% CI [0.932, 7.136].

## Discussion

This cross-sectional clinical study focused on two parameters (bone density and mandibular angle value), capable of characterising morphological and structural variations in the mandibular bone of bruxers, using CBCT imaging. The use of CBCT allowed a precise three-dimensional analysis of the mandibular anatomical structure to be conducted, as well as volume reconstruction of the mandible. This last technique was also used in this study to verify the correct spatial positioning of the tangent lines used to calculate mandibular angles. The use of CBCT also overcame one of the main limitations identified in previous studies, which employed panoramic radiographs to investigate the mandibular morphological and structural changes associated with bruxism: the superimposition of anatomical structures inherent in the use of panoramic radiographs [25]. The use of CBCT analysis therefore offered access to new data through the study of mandibular bone density in carefully selected areas of interest. Many studies in the literature use the Hounsfield scale to determine the density of body organs. This scale is used to measure the attenuation of radiation that occurs during the CT scanning process. The extreme values of this scale range from −1000 HU for air to +3000 HU for bone. Value 0 corresponds to water [26]. A bias may be introduced into the interpretation of results, by variations in grayscale depending on how each acquisition system is calibrated: bone density values may therefore vary depending on the acquisition system and the parameters used during acquisition [27]. To compensate for this, all CBCTs were performed by the same person, with identical settings on a single X-ray machine.

Eight regions of interest distributed over the entire mandible were used (the apical zones of medial incisors 31 and 41, canines 33 and 43, first premolars 34 and 44, and the mesial roots of first molars 36 and 46). The tooth type most commonly absent was the mandibular first molar. This is in line with the data available in the literature, insofar as the first molars, because they erupt at the age of six, are the permanent teeth that are most susceptible to early-onset caries. In the event of poor patient follow-up, this damage can progress rapidly and lead to tooth loss in the medium term [28].

The results of these analyses showed a statistically significant difference between the bruxer and non-bruxer groups for mean mandibular density (all sites combined), with higher density values in bruxers than in non-bruxers. Taken individually, only the sites associated with teeth 31, 34, 43, 44 showed a statistically significant difference. Interestingly, the only site that showed a statistically significant difference in density between bruxers and non-bruxers, on both right and left, was that of the mandibular first premolar: this result is in line with previous studies carried out using panoramic radiographs [29,30]. Further, the density values found for each region of interest in our study were consistent with the indications available in the literature. Indeed, we noted a progressive decrease in bone density from the symphyseal region (where 31, 41, 33, 43 are located) to the posterior zone (molars), in line with the work of Hao *et al.* [31]. These considerations are invaluable in clinical practice, in the fields of implantology and dentofacial orthopaedics, for example. A higher mandibular density expressed in Hounsfield units suggests a more mineralized bone in bruxer patients, but potentially less vascularized [31]. This can have several biological repercussions: it can enhance the implant's primary stability but also cause bone necrosis during drilling if the speed is too high. Insertion torque also needs to be adapted to avoid excessive bone compression. Thus, a better understanding of variations in mandibular density according to patient profile would improve implant planning, both in terms of evaluating the recipient site and selecting the type of implant. In orthodontics, tooth movements may be more difficult to achieve, with potentially longer treatment times. Control of the forces applied to avoid root resorption is necessary. Knowing the bone density associated with the patient's profile would therefore enable orthodontic forces to be adapted and dental movements to be better planned (in terms of time, sequence and anchorage, for example). We consider that the methodology we employed was innovative, insofar as no other study in the international published literature has used CBCT to carry out and analyse multiple bone density measurements taken from several apical zones in dentate bruxer patients. Indeed, only one study, that of Serafim *et al.* [21], has used CBCT to study morphological variations of the mandible, in a population of adult patients undergoing orthodontic treatment, without, however, studying mandibular bone density. The use of 8 different measurements per mandible served several purposes. Firstly, it ensured the reliability of the final result, by ruling out a possible bias in the selection of the area studied: the results showed that all 8 measurements established a single trend, shared by all CBCTs in the sample, demonstrating a significant difference in density between bruxers and non-bruxers. In addition, this multi-site approach allowed a more detailed analysis to be conducted by mandibular region (symphyseal, premolar zone, molar zone), which seemed relevant in light of the current understanding of bruxism. Indeed, it is possible to imagine the hypothesis that the groups of teeth required to support the forces developed by the muscles of the manducatory apparatus differ, according to the type of bruxism identified (centric or eccentric being the most common forms) [4]. Thus, in clenching bruxism (centric), it is possible to envisage that these forces are more concentrated on the molars, due to their posterior position on the arch, which places them in the zone where the stresses induced by muscle contraction will be greatest. Conversely, in grinding bruxism (eccentric bruxism), it would be possible to envisage that the anterior teeth (incisors, canines), or even the premolars, would be more subject to muscular forces in a group function context. In a physiological situation, these teeth are responsible for lateral and propulsive movements [32,33]; on clinical examination, tooth wear due to attrition is often found to be preferentially localised to the incisal edges and the tips of canines, before affecting the entire arch. Thus, the identification of certain differences in density and their correlation with bruxism type (sleep or awake bruxism, clenching or grinding bruxism, for instance) seems a valuable future line of research to be considered; the a posteriori use of computerised medical records has not made it possible to collect sufficiently detailed information on bruxism to enable this type of analysis.

The second parameter studied was mandibular angle value in bruxers and non-bruxers using CBCTs. The results presented by our study corroborate those of Serafim *et al.* [21], which showed a decreased mandibular angle in bruxers compared to non-bruxers, on both the left and right, and those of the study of Karakis *et al.* [34] in female bruxer patients. These results are also consistent with those of a study based on panoramic radiographs [35]. Indeed, Padmaja Satheeswarakumar *et al*. [35] also found a decreased mandibular angle in bruxers. It would therefore be conceivable to consider bruxism as a potential factor in the development of a decreased lower face height. One hypothesis would be that hyperactivity of the elevator muscles, mainly masseter, could impact the mandible during its growth, resulting in a mandibular angle whose value would be decreased due to muscle contraction. This line of research, currently supported by studies in animal models [36,37], would benefit from being extended to humans. To our knowledge, this has not yet been done.

Many clinical applications could be envisaged for this second parameter. A decreased mandibular angle in bruxers compared to non-bruxers may indicate a mechanical risk factor in implantology (implant overload, loss of osseointegration or prosthetic fracture, for example). In orthodontics, it can be predictive of the response to the envisaged treatment (speed of tooth movement, risk of root resorption...).

### Study limitations and recommendations for future research

Although the use of CBCT made it possible to circumvent the bias due to superimposed anatomical elements inherent in the use of panoramic radiography, several parameters remain to be corrected. The first is the requirement, in conducting this research, to carry out a retrospective study. Indeed, by forcing researchers to rely on sometimes incomplete medical records, a more detailed analysis of certain aspects, such as bruxism subtypes or possible confounding factors (such as occlusal classification, medication use, parafunctional habits), was not possible, and no statistical adjustment was performed for age and sex. In addition, the methods used to diagnose bruxism were not necessarily detailed in the medical records retrieved, which may have impacted the reliability of subject allocation to the two patient groups assembled for this study. When data collected from a patient's computerised medical record were inconsistent, or there was doubt as to the reliability of the bruxism diagnosis, the patient was not included in the study. However, in the absence of a standardised diagnostic approach to bruxism for all patients included, the composition of the groups may be open to question. Thus, a cross-sectional observational study, enabling patients to be included over time using a validated and reproducible diagnostic method such as the STAB [38], would make it possible to continue the research initiated in this preliminary cross-sectional study.

### Conclusion

This cross-sectional study based on CBCT imaging used two parameters to evaluate the presence of morphological variations of mandibular bone in adult bruxers and non-bruxers. Differences have been shown between the two groups, with a significant difference for mean mandibular alveolar bone density (all sites combined). Higher density values were found in bruxers than in non-bruxers. Furthermore, mandibular angle values were significantly lower in bruxer group. Further studies are needed to confirm these findings and determine other consequences of bruxism on the mandible, both in terms of anatomical location and morphological modification.

### Supporting information

**S1 Appendix. STROBE Checklist.**
(PDF)

**S1 Table. Measurement table_bone density.**
(CSV)

**S2 Table. Measurement table for intra-operator repeatability_ bone density.**
(CSV)

**S3 Table. Measurement table for inter-operator-reproducibility_bone density.**
(CSV)

**S4 Table. Descriptive statistics for the non-bruxer group (68 mandibles).**
(PDF)

**S5 Table. Descriptive statistics for the bruxer group (45 mandibles).**
(PDF)

**S6 Table. Measurement table _mandibular angle.**
(CSV)

**S7 Table. Measurement table for intra-operator repeatability_mandibular angle.**
(CSV)

## Author contributions

**Conceptualization:** Estelle Casazza.

**Formal analysis:** Benoit Ballester.

**Investigation:** Estelle Casazza, Benoit Ballester, Clémence Vernet.

**Methodology:** Estelle Casazza.

**Project administration:** Estelle Casazza.

**Resources:** Estelle Casazza.

**Supervision:** Anne Raskin.

**Validation:** Anne Raskin.

**Visualization:** Camille Philip-Alliez.

**Writing – original draft:** Estelle Casazza.

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
