## [Decision Letter · Decision Letter 0]

6 Aug 2025

Dear Dr. CASAZZA,

Thank you for submitting your manuscript to PLOS ONE. After careful consideration, we feel that it has merit but does not fully meet PLOS ONE’s publication criteria as it currently stands. Therefore, we invite you to submit a revised version of the manuscript that addresses the points raised during the review process.

We look forward to receiving your revised manuscript.

Kind regards,

Cristiano Miranda de Araujo

Academic Editor

PLOS ONE

Journal Requirements:

3. We note that Figures 1 and 2 in your submission contain copyrighted images. All PLOS content is published under the Creative Commons Attribution License (CC BY 4.0), which means that the manuscript, images, and Supporting Information files will be freely available online, and any third party is permitted to access, download, copy, distribute, and use these materials in any way, even commercially, with proper attribution. For more information, see our copyright guidelines: http://journals.plos.org/plosone/s/licenses-and-copyright.

1. You may seek permission from the original copyright holder of Figures 1 and 2 to publish the content specifically under the CC BY 4.0 license.

4. We note that there is identifying data in the Supporting Information files S2_Table.csv, S3_Table.csv, S4_Table.csv, S5_Table.csv, and S6_Table.csv. Due to the inclusion of these potentially identifying data, we have removed this file from your file inventory. Prior to sharing human research participant data, authors should consult with an ethics committee to ensure data are shared in accordance with participant consent and all applicable local laws.

-Location data

Data that are not directly identifying may also be inappropriate to share, as in combination they can become identifying. For example, data collected from a small group of participants, vulnerable populations, or private groups should not be shared if they involve indirect identifiers (such as sex, ethnicity, location, etc.) that may risk the identification of study participants

Please remove or anonymize all personal information (patient IDs), ensure that the data shared are in accordance with participant consent, and re-upload a fully anonymized data set. Please note that spreadsheet columns with personal information must be removed and not hidden as all hidden columns will appear in the published file.

Reviewers' comments:

Reviewer's Responses to Questions

**Comments to the Author**

1. Is the manuscript technically sound, and do the data support the conclusions?

Reviewer #1: Yes

Reviewer #2: Yes

2. Has the statistical analysis been performed appropriately and rigorously?

Reviewer #1: Yes

Reviewer #2: Yes

3. Have the authors made all data underlying the findings in their manuscript fully available?

Reviewer #1: Yes

Reviewer #2: Yes

4. Is the manuscript presented in an intelligible fashion and written in standard English?

Reviewer #1: Yes

Reviewer #2: Yes

Reviewer #1: This manuscript presents a well-conducted cross-sectional study investigating morphological differences in the mandibles of bruxer and non-bruxer adults using CBCT. The use of both 3D Slicer® and Romexis® software for assessing bone density and mandibular angle measurements strengthens the methodological rigor. The statistical analyses appear appropriate, and the discussion is comprehensive, connecting the findings to relevant literature.

However, before this manuscript is suitable for publication, I recommend the following revisions:

1. Introduction and Literature Review

The introduction could benefit from a clearer identification of the gap in the literature. While panoramic radiographs have been used previously, the novelty and clinical relevance of employing CBCT in this specific context should be more explicitly stated.

The definition of bruxism is accurate, but a brief clarification on the difference between awake and sleep bruxism—and whether this distinction was considered—would enhance the reader’s understanding.

2. Methodological Clarifications

Diagnostic Criteria for Bruxism: One of the main limitations of the study is the absence of a standardized diagnostic protocol for identifying bruxers. Please clarify whether diagnosis was based solely on clinical records, self-report, clinical signs (e.g., tooth wear), or validated instruments. If diagnosis was retrospective and inconsistent, this should be clearly emphasized as a limitation.

Sample Size Calculation: Although a sample size calculation is presented, it is unclear whether it was based on the primary outcome measures (bone density or mandibular angle). Please specify the statistical parameters and assumptions used.

3. Discussion and Interpretation

Clinical implications of the findings—particularly regarding implant planning, orthodontics, or risk of mandibular changes—should be more explicitly discussed.

The association between decreased mandibular angle and bruxism is noteworthy. Please elaborate on whether this may reflect muscular hyperactivity, skeletal remodeling, or growth-related factors.

The lack of control for potential confounding variables (e.g., sex, occlusal classification, medication use, parafunctional habits) should be acknowledged more directly.

Reviewer #2: 1. Sample size calculation:

The sample size was calculated based on prevalence estimation (22%) and a 7% margin of error, which is not appropriate for a comparative study involving continuous variables. I recommend recalculating the sample size based on:

-A clinically relevant minimum difference between groups;

-An estimated standard deviation (from pilot data or literature);

-A significance level of α = 0.05;

-Statistical power of at least 80% (1-β ≥ 0.8).

2. Bruxism diagnosis:

I suggest clarifying the criteria used to classify subjects as bruxers or non-bruxers, as recorded in the medical charts. The absence of standardized or validated diagnostic tools (such as STAB) introduces classification bias, which should be clearly acknowledged as a limitation. If possible, I encourage providing descriptive data on bruxism subtypes (sleep vs. awake, centric vs. eccentric), which could help strengthen the clinical interpretation.

3. Confounding factors:

There are apparent age differences between groups, yet no statistical adjustment was performed. I recommend using multivariate models (e.g., linear regression) or stratified analysis to control for potential confounders such as age and sex. If not feasible, this limitation should be explicitly acknowledged in the discussion.

4. Excessive number of tables and data simplification:

The manuscript includes a large number of tables, many of which present redundant or overly granular data. I suggest consolidating tables where possible, grouping by anatomical region (e.g., symphyseal, premolar, molar areas), and simplifying the presentation to improve clarity.

5. Clinical relevance of findings:

Beyond statistical significance, I suggest discussing whether the observed differences (in HU and degrees) are clinically meaningful and how they might influence treatment planning in implantology or orthodontics.

6. Interpretation and speculation:

The discussion raises interesting hypotheses regarding the relationship between bruxism type and anatomical site affected. However, these associations were not directly assessed in this study. I recommend clearly stating that these are hypothetical implications, to be explored in future prospective research.

7. Age stratification (≤45 vs. >45 years):

If the division by age group was applied in the tables or analysis, I suggest justifying the rationale for this cutoff. If no statistical comparison was performed based on age, this stratification should be removed or explained as purely descriptive.

The study addresses an important topic and employs a robust imaging methodology. With improvements in the methodological rigor, statistical analysis, and data presentation, I believe this manuscript may offer a valuable contribution to the field of dental imaging and bruxism research.

**Do you want your identity to be public for this peer review?** For information about this choice, including consent withdrawal, please see our Privacy Policy

Reviewer #1: No

Reviewer #2: No

---

## [Author Response · Author response to Decision Letter 1]

28 Aug 2025

Academic Editor

Dear Editor,

Thank you for your review of our manuscript.

We have modified the items you mentioned in your feedback. You will find our responses to each of your suggestions in blue.

Kind regards,

The authors

The requested checks have been carried out, and changes made to figure legends and supporting information.

In accordance with French regulations, single-center retrospective studies using pre-existing data for research purposes do not require written consent from patients, provided they have been informed in advance and do not object, and subject to a favorable opinion from an ethics committee.

This indication has been added to the manuscript (lines 99-100).

3. We note that Figures 1 and 2 in your submission contain copyrighted images. All PLOS content is published under the Creative Commons Attribution License (CC BY 4.0), which means that the manuscript, images, and Supporting Information files will be freely available online, and any third party is permitted to access, download, copy, distribute, and use these materials in any way, even commercially, with proper attribution.

• Figure 1 is based on a screenshot from the 3D Slicer software, which is a free open source software distributed under a BSD style license. The license does not impose restrictions on the use of the software, as indicated on the software website at the following address: https://www.slicer.org/commercial-use.html

This figure has therefore been retained in the manuscript.

• Figure 2 has been removed from the manuscript.

4. We note that there is identifying data in the Supporting Information files S2_Table.csv, S3_Table.csv, S4_Table.csv, S5_Table.csv, and S6_Table.csv. Due to the inclusion of these potentially identifying data, we have removed this file from your file inventory. Prior to sharing human research participant data, authors should consult with an ethics committee to ensure data are shared in accordance with participant consent and all applicable local laws.

Thank you for these clarifications. The necessary modifications have been made.

Reviewers

Dear Reviewers,

Thank you for your careful review of the manuscript and your suggestions for improving its quality.

We have considered every one of your comments to improve this article. You will find our responses to your feedback in blue.

Kind regards,

The authors

Reviewer #1

1. Introduction and Literature Review

The introduction could benefit from a clearer identification of the gap in the literature. While panoramic radiographs have been used previously, the novelty and clinical relevance of employing CBCT in this specific context should be more explicitly stated.

Thank you for your comment. It has been added (lines 79–82). In addition, this is developed in the discussion (lines 256-260).

The definition of bruxism is accurate, but a brief clarification on the difference between awake and sleep bruxism—and whether this distinction was considered—would enhance the reader’s understanding.

We have taken your suggestion into account to complete the manuscript (lines 59-60). As this distinction could not be taken into account due to the absence of this data in a certain number of medical records, it could not be exploited in this study. This element has been added to the discussion (line 313).

2. Methodological Clarifications

Diagnostic Criteria for Bruxism: One of the main limitations of the study is the absence of a standardized diagnostic protocol for identifying bruxers. Please clarify whether diagnosis was based solely on clinical records, self-report, clinical signs (e.g., tooth wear), or validated instruments. If diagnosis was retrospective and inconsistent, this should be clearly emphasized as a limitation.

Thank you for your comment. This was clarified in the inclusion criteria (lines 106-108).

Sample Size Calculation: Although a sample size calculation is presented, it is unclear whether it was based on the primary outcome measures (bone density or mandibular angle). Please specify the statistical parameters and assumptions used.

Thank you for your comment. The requested information has been added (lines 125-129).

3. Discussion and Interpretation

Clinical implications of the findings—particularly regarding implant planning, orthodontics, or risk of mandibular changes—should be more explicitly discussed.

Your comments have been taken into account, and the clinical implications are detailed lines 283-292 and 326-329.

The association between decreased mandibular angle and bruxism is noteworthy. Please elaborate on whether this may reflect muscular hyperactivity, skeletal remodeling, or growth-related factors.

These elements have been mentioned lines 322-325.

The lack of control for potential confounding variables (e.g., sex, occlusal classification, medication use, parafunctional habits) should be acknowledged more directly.

Thank you for your comment. Details of confounding variables have been added line 336.

Reviewer #2

1. Sample size calculation:

The sample size was calculated based on prevalence estimation (22%) and a 7% margin of error, which is not appropriate for a comparative study involving continuous variables. I recommend recalculating the sample size based on:

-A clinically relevant minimum difference between groups;

-An estimated standard deviation (from pilot data or literature);

-A significance level of α = 0.05;

-Statistical power of at least 80% (1-β ≥ 0.8).

Thank you for this comment and the associated information. The calculation of the number of subjects required has been corrected according to your indications. It is based on an expected difference between groups of 200HU, with a standard deviation of 250HU, an alpha risk of 0.05% and a power of 90% (lines 125-129).

However, in order to anticipate the potential loss of data (missing teeth, missing mandibular angle), a number greater than the minimum size required was included, to guarantee sufficient statistical power for both parts of the study, particularly for the mandibular angle section.

2. Bruxism diagnosis:

I suggest clarifying the criteria used to classify subjects as bruxers or non-bruxers, as recorded in the medical charts. The absence of standardized or validated diagnostic tools (such as STAB) introduces classification bias, which should be clearly acknowledged as a limitation. If possible, I encourage providing descriptive data on bruxism subtypes (sleep vs. awake, centric vs. eccentric), which could help strengthen the clinical interpretation.

Thank you for your comment. Additional descriptive elements concerning the diagnosis of bruxism have been added lines 106-108.

Unfortunately, data on bruxism subtypes were not sufficiently mentioned in the medical records to allow clinical interpretations. This was discussed at the end of the manuscript.

3. Confounding factors:

There are apparent age differences between groups, yet no statistical adjustment was performed. I recommend using multivariate models (e.g., linear regression) or stratified analysis to control for potential confounders such as age and sex. If not feasible, this limitation should be explicitly acknowledged in the discussion.

Thank you for your comment. The corresponding change has been added (line 337).

4. Excessive number of tables and data simplification:

The manuscript includes a large number of tables, many of which present redundant or overly granular data. I suggest consolidating tables where possible, grouping by anatomical region (e.g., symphyseal, premolar, molar areas), and simplifying the presentation to improve clarity.

Thank you for your comment.

The presentation of the results has been modified (lines 198-209). To improve readability and avoid repetition, 2 tables have been removed from the manuscript, and are now part of the supporting information.

We preferred to keep the values in a single table to give the reader an overview of the results. We hope the result is more in line with your expectations.

5. Clinical relevance of findings:

Beyond statistical significance, I suggest discussing whether the observed differences (in HU and degrees) are clinically meaningful and how they might influence treatment planning in implantology or orthodontics.

Thank you for your comment. Modifications has been added lines 282-292 and 326-329.

7. Age stratification (≤45 vs. >45 years):

If the division by age group was applied in the tables or analysis, I suggest justifying the rationale for this cutoff. If no statistical comparison was performed based on age, this stratification should be removed or explained as purely descriptive.

Thank you for your comment. The division by age group was purely descriptive. The cut-off of 45 years old was the median age of the patients included in the study. This clarification has been added to the manuscript (line 189).

---

## [Decision Letter · Decision Letter 1]

8 Oct 2025

Dear Dr. CASAZZA,

Thank you for submitting your manuscript to PLOS ONE. After careful consideration, we feel that it has merit but does not fully meet PLOS ONE’s publication criteria as it currently stands. Therefore, we invite you to submit a revised version of the manuscript that addresses the points raised during the review process.

We look forward to receiving your revised manuscript.

Kind regards,

Cristiano Miranda de Araujo

Academic Editor

PLOS ONE

Journal Requirements:

Reviewers' comments:

Reviewer's Responses to Questions

**Comments to the Author**

Reviewer #1: All comments have been addressed

Reviewer #2: (No Response)

2. Is the manuscript technically sound, and do the data support the conclusions?

Reviewer #1: Yes

Reviewer #2: Yes

3. Has the statistical analysis been performed appropriately and rigorously?

Reviewer #1: Yes

Reviewer #2: Yes

4. Have the authors made all data underlying the findings in their manuscript fully available?

Reviewer #1: Yes

Reviewer #2: Yes

5. Is the manuscript presented in an intelligible fashion and written in standard English?

Reviewer #1: Yes

Reviewer #2: Yes

Reviewer #1: All comments are completely answered. Thank you for opportunity to review. This is a very good study. The authors are to be congratulated for their research and should feel encouraged to increase the sample size in future investigations

Reviewer #2: Overall, the manuscript has been improved, and most of the limitations highlighted in the previous review have been adequately addressed or properly justified. The study presents relevant findings, but I suggest a few additional revisions before acceptance:

-Review of references

The citation Lumetti S. et al., 2016 [9] does not adequately support the statement in lines 57–58: “can weaken the teeth and the temporomandibular joints, and generate mastication myalgia [9]”. Please revise this reference and consider replacing it with a more appropriate source.

-Study limitations

Although the limitations are discussed, I recommend moving the content currently found in lines 316–329 into the Discussion as the final paragraph. Alternatively, a specific subsection entitled “Study limitations and recommendations for future research” could be created.

-Conclusion

Please ensure that the Conclusion is concise and directly answers the research question: whether differences exist between bruxers and non-bruxers, and in which variables these differences were found.

**Do you want your identity to be public for this peer review?** For information about this choice, including consent withdrawal, please see our Privacy Policy

Reviewer #1: No

Reviewer #2: No

---

## [Author Response · Author response to Decision Letter 2]

14 Oct 2025

Reviewer #1

Reviewer #1: All comments are completely answered. Thank you for opportunity to review. This is a very good study. The authors are to be congratulated for their research and should feel encouraged to increase the sample size in future investigations.

We would like to thank you for your feedback and the valuable advice you gave us to improve the quality of the manuscript.

Reviewer #2

Overall, the manuscript has been improved, and most of the limitations highlighted in the previous review have been adequately addressed or properly justified. The study presents relevant findings, but I suggest a few additional revisions before acceptance:

-Review of references

The citation Lumetti S. et al., 2016 [9] does not adequately support the statement in lines 57–58: “can weaken the teeth and the temporomandibular joints, and generate mastication myalgia [9]”. Please revise this reference and consider replacing it with a more appropriate source.

Thank you for your comment. This citation has been replaced by a more appropriate reference.

Matusz K, Maciejewska-Szaniec Z, Gredes T, Pobudek-Radzikowska M, Glapiński M, Górna N, Przystańska A. Common therapeutic approaches in sleep and awake bruxism - an overview. Neurol Neurochir Pol. 2022;56(6):455-463. doi: 10.5603/PJNNS.a2022.0073. Epub 2022 Nov 29.

-Study limitations

Although the limitations are discussed, I recommend moving the content currently found in lines 316–329 into the Discussion as the final paragraph. Alternatively, a specific subsection entitled “Study limitations and recommendations for future research” could be created.

Thanks to your suggestion, a section “Study limitations and recommendations for future research” has been created at the end of the discussion (line 315).

-Conclusion

Please ensure that the Conclusion is concise and directly answers the research question: whether differences exist between bruxers and non-bruxers, and in which variables these differences were found.

Thank you for your comment. The conclusion has been rewritten according to your instructions (lines 333-339).

---

## [Decision Letter · Decision Letter 2]

19 Jan 2026

Dear Dr. CASAZZA,

Thank you for submitting your manuscript to PLOS ONE. After careful consideration, we feel that it has merit but does not fully meet PLOS ONE’s publication criteria as it currently stands. Therefore, we invite you to submit a revised version of the manuscript that addresses the points raised during the review process.

The manuscript has improved. A minor revision is still suggested by Reviewer 2, and we recommend that the authors carefully observe this suggestion.

We look forward to receiving your revised manuscript.

Kind regards,

Sergio Luiz Mota-Junior, Ph.D.

Academic Editor

PLOS One

Journal Requirements:

Additional Editor Comments:

The manuscript has improved. However, a minor revision is still suggested by Reviewer 2.

Reviewers' comments:

Reviewer's Responses to Questions

**Comments to the Author**

Reviewer #1: All comments have been addressed

Reviewer #2: All comments have been addressed

2. Is the manuscript technically sound, and do the data support the conclusions?

Reviewer #1: Yes

Reviewer #2: Yes

3. Has the statistical analysis been performed appropriately and rigorously?

Reviewer #1: Yes

Reviewer #2: Yes

4. Have the authors made all data underlying the findings in their manuscript fully available?

Reviewer #1: Yes

Reviewer #2: Yes

5. Is the manuscript presented in an intelligible fashion and written in standard English?

Reviewer #1: Yes

Reviewer #2: Yes

Reviewer #1: Thank you for all corrections and good luck in your research. It’s a very complex and necessary research

Reviewer #2: I congratulate the authors once again on the development of this work. All recommendations have been accepted and addressed. During the process of reviewing this article, an updated concept of bruxism was published. I simply recommend that the new concept be incorporated into your work. The reference is as follows: Verhoeff MC, Lobbezoo F, Ahlberg J, Bender S, Bracci A, Colonna A, Dal Fabbro C, Durham J, Glaros AG, Häggman-Henrikson B, Kato T, Koutris M, Lavigne GJ, Nykänen L, Raphael KG, Svensson P, Wieckiewicz M, Manfredini D. Updating the Bruxism Definitions: Report of an International Consensus Meeting. J Oral Rehabil. 2025 Sep;52(9):1335-1342. doi: 10.1111/joor.13985. Epub 2025 May 1. PMID: 40312776; PMCID: PMC12408978.

**Do you want your identity to be public for this peer review?** For information about this choice, including consent withdrawal, please see our Privacy Policy

Reviewer #1: No

Reviewer #2: No

---

## [Author Response · Author response to Decision Letter 3]

19 Jan 2026

Reviewer #1: Thank you for all corrections and good luck in your research. It’s a very complex and necessary research

We would like to thank you for your feedback to improve the quality of the manuscript.

Reviewer #2: I congratulate the authors once again on the development of this work. All recommendations have been accepted and addressed. During the process of reviewing this article, an updated concept of bruxism was published. I simply recommend that the new concept be incorporated into your work. The reference is as follows: Verhoeff MC, Lobbezoo F, Ahlberg J, Bender S, Bracci A, Colonna A, Dal Fabbro C, Durham J, Glaros AG, Häggman-Henrikson B, Kato T, Koutris M, Lavigne GJ, Nykänen L, Raphael KG, Svensson P, Wieckiewicz M, Manfredini D. Updating the Bruxism Definitions: Report of an International Consensus Meeting. J Oral Rehabil. 2025 Sep;52(9):1335-1342. doi: 10.1111/joor.13985. Epub 2025 May 1. PMID: 40312776; PMCID: PMC12408978

Thank you for your comment and your interest in this work. This citation has been added to update the bibliography.

---

## [Editor Report · Decision Letter 3]

25 Jan 2026

Evaluation of morphological variations of mandibular bone in adult bruxers using CBCT: a cross-sectional study

PONE-D-25-25414R3

Dear Dr. Casazza,

We’re pleased to inform you that your manuscript has been judged scientifically suitable for publication and will be formally accepted for publication once it meets all outstanding technical requirements.

Kind regards,

Sergio Luiz Mota-Junior, Ph.D.

Academic Editor

PLOS One

Additional Editor Comments (optional):

Dear Authors,

Thank you for submitting the revised version of your manuscript to PLOS ONE.

We have now completed the evaluation of the revised manuscript and the responses to the reviewers’ comments. Based on this assessment, the manuscript is adequate for publication in PLOS One.

---

## [Editor Report · Acceptance letter]

PONE-D-25-25414R3

PLOS One

Dear Dr. CASAZZA,

I'm pleased to inform you that your manuscript has been deemed suitable for publication in PLOS One. Congratulations! Your manuscript is now being handed over to our production team.

Kind regards,

on behalf of

Dr. Sergio Luiz Mota-Junior

Academic Editor

PLOS One